# An agent-based model on antimicrobial de-escalation in intensive care units: Implications on clinical trial design

Xi Huo[1]*, Ping Liu[2]

**1** Department of Mathematics, University of Miami, Coral Gables, FL, United States of Ameica, **2** LinkedIn Corporation, Mountain View, CA, United States of Ameica

☉ These authors contributed equally to this work.

\* x.huo@math.miami.edu

**Data Availability Statement:** All coding files are available from the GitHub repository https://github.com/pliu19/ABM-Antimicrobial.

**Funding:** XH was partially supported by the National Science Foundation (DMS-1853622 and DMS-2052648) and the College of Arts and

## Abstract

Antimicrobial de-escalation refers to reducing the spectrum of antibiotics used in treating bacterial infections. This strategy is widely recommended in many antimicrobial stewardship programs and is believed to reduce patients' exposure to broad-spectrum antibiotics and prevent resistance. However, the ecological benefits of de-escalation have not been universally observed in clinical studies. This paper conducts computer simulations to assess the ecological effects of de-escalation on the resistance prevalence of *Pseudomonas aeruginosa*—a frequent pathogen causing nosocomial infections. Synthetic data produced by the models are then used to estimate the sample size and study period needed to observe the predicted effects in clinical trials. Our results show that de-escalation can reduce colonization and infections caused by bacterial strains resistant to the empiric antibiotic, limit the use of broad-spectrum antibiotics, and avoid inappropriate empiric therapies. Further, we show that de-escalation could reduce the overall super-infection incidence, and this benefit becomes more evident under good compliance with hand hygiene protocols among health care workers. Finally, we find that any clinical study aiming to observe the essential effects of de-escalation should involve at least ten arms and last for four years—a size never attained in prior studies. This study explains the controversial findings of de-escalation in previous clinical studies and illustrates how mathematical models can inform outcome expectations and guide the design of clinical studies.

## 1 Introduction

Patients admitted to intensive care units (ICUs) are vulnerable to life-threatening infections due to weakened immune systems and treatments with invasive medical devices. Thus there is always a pressing need for an immediate and effective empirical antibiotic therapy, which is crucial in reducing mortality rates, length of ICU stays, and medical costs. Broad-spectrum antibiotics are used in empiric therapy when the susceptibility of infecting pathogen is unknown, but the overuse of them are linked to the rapid development of antimicrobial

Sciences at the University of Miami. This report is solely the responsibility of the authors and does not necessarily represent the official views of the National Science Foundation and the University of Miami.

**Competing interests:** The authors have declared that no competing interests exist.

resistance. Antimicrobial *de-escalation* refers to the procedure of reducing the spectrum of antibiotics in treatments by stopping unnecessary antibiotics or switching to narrower spectrum antibiotics whenever laboratory test confirms susceptibility. Whereas the opposite strategy, named *continuation*, only recommends drug switching whenever a treatment needs to be corrected. De-escalation is considered as a remedy to mitigate patients' exposure to the broad-spectrum antibiotics during the empiric therapy, thus is assumed to reduce the development of resistance to the broad-spectrum antibiotics [1–7].

Though widely practiced, the specific benefits and trade-offs of de-escalation are still unclear and should be evaluated on both individual and ecological levels. Many clinical studies have been conducted to examine the individual-level outcomes of de-escalation in safety, length of ICU stay, mortality, and resistance development. Systematic reviews and meta-analyses have shown that de-escalation is a safe treatment strategy. And no significant difference has been observed between de-escalation and continuation on the individual patient outcomes, including length of ICU stay and mortality [8–12]. However, the ecological impacts of de-escalation on the transmission and development of resistant bacterial strains are still unknown [13]. Due to the large number of bacterial species and antibiotics, there is a lack of universal definition and protocol for de-escalation. Thus conclusions on the ecological effects varied from study to study. The prevalence of resistant bacteria is caused by the patient-level intrinsic resistance development and nosocomial transmission. Although shortening patients' exposure window to broad-spectrum antibiotics, de-escalation also exposes patients to more than one antibiotic, which might lead to the development of dual-drug resistance. In addition, the overall antibiotic pressure and the microbial ecology in the ICU could influence each other in a complicated and bidirectional fashion. Thus the ecological impacts of de-escalation are not intuitively perceivable.

Previously, differential equation models have been developed to investigate the ecological effects of de-escalation [14, 15]. Deterministic models can provide theoretical insights into ecological systems with minimal computational cost but fail to reflect the stochastic uncertainties in observations. For example, the time-dependent solution and the long-term behavior of deterministic models are definite given fixed parameter values and initial conditions. But in reality, two identical ICUs could have significantly different outcomes due to stochastic events. Further, differential equation models are primarily applicable in modeling population dynamics when assuming a large population size to ignore heterogeneity. However, given a small cohort of the patient population in ICUs, it is always hard to ignore the impact of randomness. Therefore, stochastic models are more suitable to simulate the patient and HCW activities in hospitals.

Agent-based models have been extensively used in ecology dealing with complex systems with autonomous entities [16]. Comparing to deterministic differential equation models, agent-based models possess advantages in describing events from the individual point of view and are more suitable in modeling small groups of population such as patients in ICUs. In the past decade, both types have been widely applied to investigate the effects of various antimicrobial stewardship strategies, to name a few [17–32] and we also refer to the review and references in [33, 34].

This paper develops an agent-based model to simulate the patient and HCW population in an ICU under the antimicrobial protocols of de-escalation and continuation. The model is applied to generate synthetic data of an assumed two-armed randomized controlled trial (RCT) that evaluates the ecological benefits and trade-offs of de-escalation. We first compare the simulation results with real hospital data to calibrate the model parameters. Then we perform statistical analyses to determine the theoretical effects of de-escalation. Finally, we inform the number of study arms and the length of study periods that guarantee the detectability of

the predicted outcomes. This study indicates that mathematical modeling could serve as an economic tool to envision possible clinical study outcomes, inform the proper sample size and study length, or even avoid costly and ineffective trials. The model's publicly available code provides a baseline framework as an agent-based model for the transmission of nosocomial pathogens between patients and HCWs.

## 2 Methods

We consider two types of agents in an ICU: patients receiving critical medical care and health care workers (HCWs) caring for them. Each agent (individual) has attributes whose values/status may vary with time. A complete list of bacterial species and antibiotics would differ from hospital to hospital, and incorporating all drug-bug pairs would lead to an over-complicated model. Therefore, we focus on the transmission and resistance of *Pseudomonas aeruginosa* (PA)—a primarily nosocomial pathogen with a high resistance development rate—and categorize all non-*Pseudomonas* (non-PA) bacteria into one single group. All simulations and figures are programmed and generated in *Python*, with main codes available online: https://github.com/pliu19/ABM-Antimicrobial.

### 2.1 Model assumptions

**Baseline set-up for the ICU.**   We consider an ICU with a fixed patient-HCW ratio of 4:1. Each day consists of 3 shifts and each lasting 8 hours. Each patient receives routine care once during every shift. Each HCW works one shift per day and treats four patients per shift. Contacts between patients and HCWs only happen during the routine visits, when nosocomial transmissions could occur.

**Antibiotics and spectrums.**   We focus on the evolution and transmission of PA in the ICU. Thus we firstly subdivide pathogens circulating in the ICU into PA species subject to different resistance profiles and other non-PA species. Piperacillin-tazobactam is often used for empiric therapy of severe infections in the ICU due to its good coverage of common pathogens. We thus consider a general use of piperacillin-tazobactam for empiric therapy. Ciprofloxacin is usually prescribed to treat a wide variety of infections and has high resistance rate in PA, we therefore consider it as the de-escalated treatment option for PA infections in definitive therapy. We refer the de-escalated antibiotic options for non-PA infections as non-pseudomonal (non-PA) drugs. Our model excludes the possibility of a pandrug-resistant bacterial strain and we assume there are last-resort drugs such as carbapenem or aminoglycoside to which no resistance would exist in the ICU. In the following, we adopt the widely used acronyms provided by the American Society for Microbiology to represent the aforementioned antibiotics: TZP for piperacillin-tazobactam and CIP for ciprofloxacin. The last-resort drugs refer to several antibiotics, and we represent such drugs by IPM for simplicity, where IPM is the abbreviation of imipenem—an antibiotic from the carbapenem class.

The PA species is further divided into several strains with respect to their resistance profile: the susceptible strain, CIP-resistant strain, TZP-resistant strain, and the dual-resistant strain. Note that all PA strains are not susceptible to non-PA drugs but are susceptible to the last-resort drugs. Fig 1(d) provides a summary of the drug spectrum information.

**Attributes of agents.**   HCWs have only one attribute about their status of *contaminated pathogens* and could contaminate more than one pathogen at the same time.

We define eight attributes for each patient: (1) *time in ICU* tracks the length of ICU stay, which increments with respect to real-time until discharge or death; (2) *infection status* of a patient can be either uninfected or infected; (3) *treatment time* records the time since empiric treatment initiation for an infected patient; (4) *drug use* denotes the antibiotic administered at

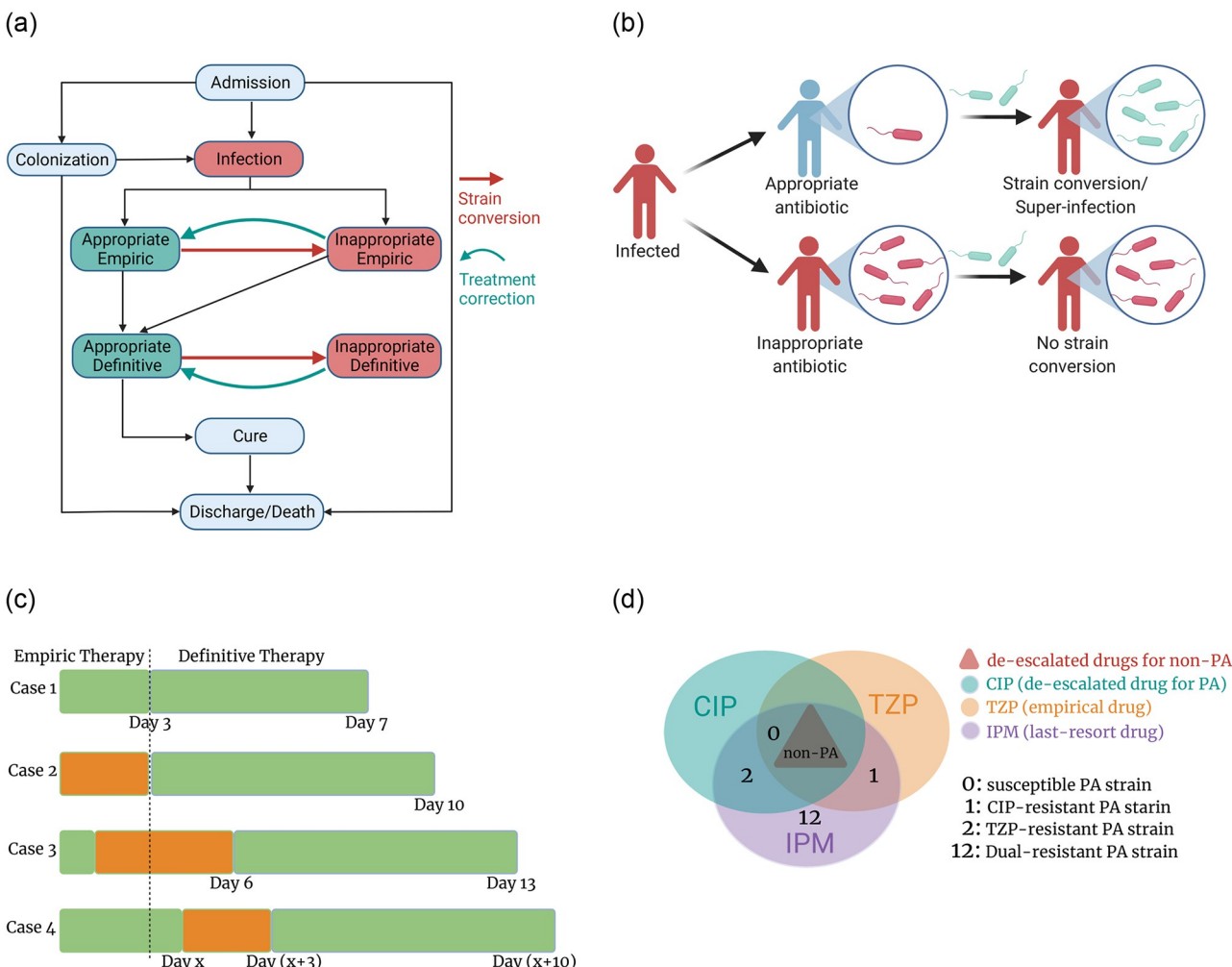

**Fig 1. Graphic illustrations of model assumptions.** (a) Assumptions on transitions of patients' infection and treatment status. (b) Appropriate antibiotics could help reduce the patient's bacteria population, hence alleviating the infection. But if the patient contracts another bacterial strain resistant to the current antibiotic, a strain switch becomes possible due to selective pressure. On the other hand, strain switch is not possible for patients under inappropriate antibiotics. (c) Case 1 represents an ideally successful treatment where a successful 4-day definitive therapy follows a successful 3-day empiric therapy. Case 2 illustrates an inappropriate 3-day empiric therapy followed with a 7-day correction of definitive therapy. Case 3 refers to a possible occurrence of strain switch during a successful empiric therapy. The unknown strain switch would result in an ineffective definitive therapy for three days, followed by a 7-day correction. Case 4 corresponds to a possible strain switch during the definitive therapy. This unknown strain switch would lead to an ineffective definitive therapy for three days followed by a 7-day correction therapy. (d) Pathogens lying inside each circle are susceptible to the corresponding antibiotic, whereas those outside the circle are resistant to the antibiotic. An infection of any non-PA species can be treated by the correspondingly de-escalated non-PA antibiotics and by the three PA-targeted antibiotics because of their relatively broad spectrum. Bacteria develop resistance against each antibiotic in specific ways. Thus the CIP-resistant strain is susceptible to TZP and *vice versa*. We assume that the last-resort drugs can ultimately treat the dual-resistant strain.

the time for an infected patient; (5) *conversion time* tracks the time since an unknown pathogen switch during the antibiotic treatment of an infected patient; (6) *dominant pathogen* labels the pathogen that dominates the current colonization or infection of a patient; (7) *lab result* denotes the dominant pathogen that causes the initial infection of a patient; (8) *super-infection* denotes whether or not the infected patient is experiencing a super-infection.

The attributes of each agent are updated accordingly to the following model events. Fig 1(a) provides a graphic illustration of the potential status change for individual patients. A detailed and complete algorithm can be found in the S1 Appendix.

**Model Event 1.** *admission, discharge, and death.* Upon admission, each patient is randomly assigned with a colonization pathogen which is reflected in the *dominant pathogen* attribute. In addition, patients not colonized with PA are assigned with a prior exposure history to PA antibiotics based on specific probabilities. Patients colonized with non-PA species and with prior exposure to PA antibiotics are likely to be colonized with a PA strain via contacts with HCWs. Whereas a PA colonization is unlikely to happen to those with no recent PA antibiotic exposures. The hazard rates of discharge and death of a patient are dependent on *time in ICU* and *infection status*. The baseline hazard functions for discharge and deaths about *time in ICU* are parameterized according to real data in [35] as sketched in Fig 2. Uninfected patients would be discharged or die with respect to the baseline hazard functions. Infected patients are less likely to be discharged with a hazard ratio of discharge $\kappa_\mu < 1$. Infected patients receiving ineffective antibiotic treatments may experience a higher death rate with a hazard ratio of death $\kappa_\nu > 1$. Upon an event of discharge or death, the patient will be immediately replaced by a newly admitted patient.

**Model Event 2.** *HCW contamination.* HCWs do not carry any strain at the beginning of their shifts. They could contaminate with PA strains during any visit to a colonized/infected patient at a probability of $q$. After taking care of a patient, an HCW can be de-contaminated if complying with the hand hygiene guidelines at a probability of 50%. In reality, HCWs are the critical agents of spreading other antibiotic-resistant pathogens, such as *methicillin-resistant Staphylococcus aureus* (MRSA) and *vancomycin-resistant enterococci* (VRE). These pathogens are grouped as non-PA species in our model and are susceptible to PA antibiotics. Therefore, their transmission would not impact the prevalence of resistance against PA antibiotics. So we simplify our assumption by ignoring the transmission of non-PA species by HCWs.

**Model Event 3.** *PA transmission.* Both colonized and infected patients with PA can spread their bacteria to HCWs. On the other hand, HCWs could transmit PA to a patient either when the patient is uninfected and has prior exposures to antimicrobials or when a super-infection (see Model Event 7) happens. Further, HCWs could pass a specific PA strain to a patient infected by another PA strain under selective pressure (see Model Event 5). We assume the probability for an HCW to contaminate a PA strain upon contact as $q$, and assume the probability of an HCW passing PA to a patient upon contact as $p$. When an HCW carries more than

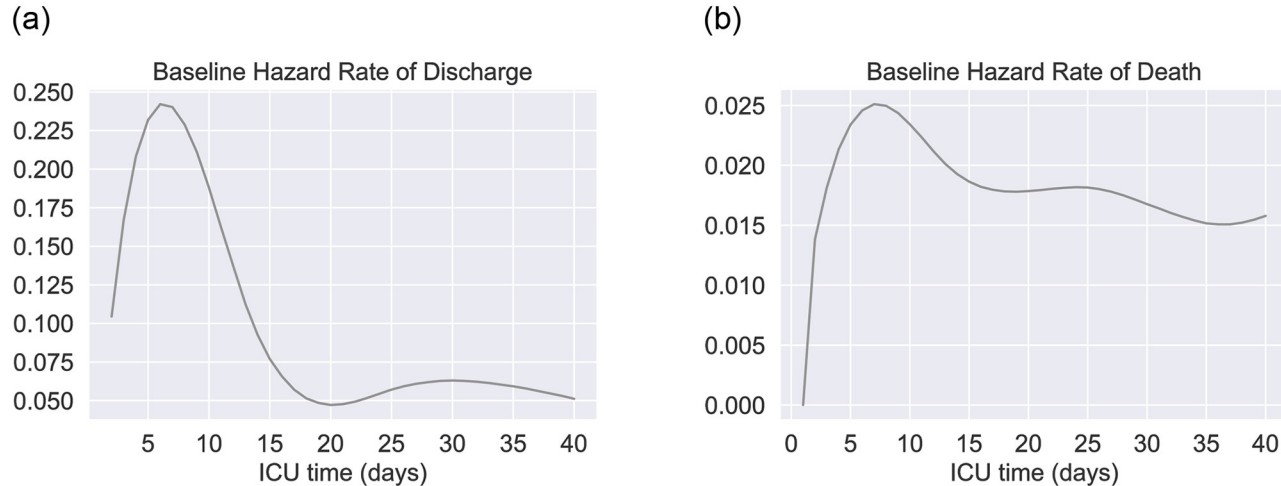

**Fig 2. Baseline hazard rate functions.** Both functions are parameterized from real data collected in [35]. (a) Hazard rate of discharge., (b) Hazard rate of death.

one strain, the patient will have an equal chance to acquire any strain the HCW carries during their contact. That is, if the HCW carries $n$ strains, the patient will have a probability $p/n$ to be colonized by each strain. No colonization would occur to uninfected patients with no prior antimicrobial exposures or uninfected patients already colonized with PA strains.

**Model Event 4.** *infection development.* Patients colonized with PA strains will have a probability $\sigma_c$ to develop infections within five days, and patients colonized with non-PA pathogens will develop infections at another probability $\sigma_x$. Infected patients would receive empiric treatments with TZP for three days, and meanwhile, their samples are sent for laboratory testing to inform definitive therapy.

**Model Event 5.** *strain conversion.* Patients infected with a PA strain could have their dominant strains converted from susceptible to resistant due to antibiotic selective pressure and intrinsic mutation. On the patient level, antibiotic pressure promotes the growth of resistant strains. Thus patients could experience a strain conversion whenever taking an antibiotic that treats the dominant strain but fails to treat the new invading strain brought by HCWs during visits. A graphic illustration about strain switch due to selective pressure is shown in Fig 1(b). Intrinsic mutation against an antibiotic would occur at a hazard rate of $\varepsilon$ per day to patients treated by the corresponding antibiotic. Dual-resistance mutation could develop at the same rate when a patient infected by the CIP-resistant strain receives TZP, or *vice versa*.

**Model Event 6.** *super-infection.* Super-infections refer to PA infections that occur after or on top of an earlier non-PA infection. Super-infections are direct results of antibiotic pressure and often occur following treatments with broad-spectrum antibiotics. The mechanisms of super-infection are similar to that of strain conversion which can be found in Fig 1(b). Therefore, patients treated for non-PA infections by TZP might be super-infected with TZP-resistant or dual-resistant PA strains upon contacts with contaminated HCWs. Similarly, patients treated by a de-escalated non-PA antibiotic might also develop super-infections with any PA strain. The probability for a super-infection to occur is relatively smaller than the probability of PA transmission, and we denote the super-infection probability as $s$.

**Model Event 7.** *treatment strategy.* Infected patients are treated empirically with TZP until phenotypic resistance testing confirms the identity and susceptibilities of the infecting pathogen. We assume testing takes three days. Thus definitive therapy starts three days after the initial infection. De-escalation and continuation only differ in how to administer definitive therapies. In the de-escalation scenario, patients will receive a definitive antibiotic with the least possible spectrum covering their initial infecting pathogen. In the continuation scenario, patients continue to receive TZP unless the testing results confirm resistance. Under both strategies, patients initially infected by TZP-resistant strain will have to switch to CIP. Likewise, patients initially infected by the dual-resistant strain will have to switch to the last-resort antibiotics. Whereas patients initially infected by non-PA pathogens or susceptible PA strain would end up with different choices of antibiotics during their definitive therapy under the two strategies.

**Model Event 8.** *treatment correction and completion.* Effective treatment for a consecutive seven days is sufficient to cure infection (case 1 in Fig 1(c)). However, empiric therapy could be inadequate from the beginning and will be corrected at the definitive stage (case 2 in Fig 1 (c)). Moreover, empiric antibiotics could fail in the middle of the therapy due to strain conversion or super-infection. Since the phenotypic testing sample would only reflect the initial infecting pathogen, the definitive therapy may remain ineffective. We assume such ineffective definitive therapy would then be corrected three days later (case 3 in Fig 1(c)). Further, definitive antibiotics could also fail in the middle of the therapy, and we assume that it will be corrected after three days (case 4 in Fig 1(c)). To avoid prolonged infections, patients under ineffective CIP or TZP treatments will receive the last-resort antibiotics for treatment correction. Patients under ineffective non-PA antibiotics will take TZP for treatment correction.

## 2.2 Parameters and calibration

We assume the probability for a newly admitted patient to have recent history of prescribing
PA antibiotics is 0.6. And we assume such a patient would have a further 0.1 probability of
being already colonized by a PA strain. Additionally, there is a 23% chance for the colonization
being resistant to TZP and 35% being resistant to CIP [36]. We assume 50% HCWs would
comply with the hand hygiene guidelines for decontamination after each visit. The hazard
rates of infection development, death, and discharge are all parameterized from the literature.
The possibility of intrinsic resistance development on the human population level is hard to
infer from the literature. We assume it to be 0.03, which agrees with the detailed discussion in
[37] based on data in [38]. Further, we assume the super-infection probability to be as low as
0.015 for patients treating with a narrow-spectrum antibiotic. We list the baseline values of all
fixed parameters in Table 1.

We calibrate the transmission probabilities between patients and HCWs ($p$ and $q$) from real
data. Specifically, we search for reasonable parameter values so that the simulation outcomes
match the data known from previous studies. To do so, we generate 100 parameter pairs of $q$
and $p$ with each value ranging from 0.05 to 0.5. We perform simulations 1000 times for each
pair for both de-escalation and control models. We then identify those parameter pairs with all
outcome measures falling into the following ranges: resistance to CIP and TZP not exceeding
70% and 50%, respectively; PA colonization prevalence ranges from 6% to 32%. These ranges
help rule out the pairs where both parameters are extremely large or small. Fig 3 summarizes
the fractions of outcome measures that fall in the credible ranges for each pair. We perform
further simulations with a selection of six qualified pairs to study our problem under different
transmission intensities: **high transmission intensity** where the mean PA colonization preva-
lence is close to 28% ($p = q = 0.3$, $p = 0.5$, $q = 0.2$ and $p = 0.2$, $q = 0.5$), and the **low transmis-
sion intensity** with that close to 12% ($p = q = 0.15$, $p = 0.5$, $q = 0.05$ and $p = 0.05$, $q = 0.5$).

## 2.3 Outcome measurements

For each simulation, we track the number of daily events and measure the incidence propor-
tions (*i.e.* cumulative events over total admission) of several important events: (i) incidence of
infections caused by each PA strain—this includes both initial infections and strain

**Table 1. Model parameters with baseline values.** In Experiment 1, all parameters are fixed at the baseline values. In
Experiment 2, each parameter is sampled from a truncated normal distribution $\mathcal{N}(\mu, \sigma)$ with $\mu$ being its baseline value
and $\sigma = 0.1\mu$.

| Parameter | Baseline Value | Reference |
|---|---|---|
| Percent of patients admitted with prior exposure to PA antibiotics ($m$) | 60% | [37] |
| Percent of patients admitted with PA colonization ($a$) | 10% | [14] |
| Percent of patients admitted with colonization of CIP-resistant strain ($r_1$) | 35% | [36] |
| Percent of patients admitted with colonization of TZP-resistant strain ($r_2$) | 23% | [36] |
| Probability of compliance with hand hygiene ($\eta$) | 0.5 | Assumed. |
| Probability for non-PA infection development ($\sigma_x$) | 0.16 | [41] |
| Probability for PA infection development ($\sigma_c$) | 0.45 | [40] |
| Probability of super-infection under narrow-spectrum antibiotics ($s$) | 0.015 | Assumed. |
| Hazard rate of discharge for colonized patients ($\mu(\cdot)$) | | [35] |
| Hazard rate of death for colonized patients ($v(\cdot)$) | | [35] |
| Hazard ratio of discharge for infected patients ($\kappa_\mu$) | 0.74 | [47, 48] |
| Hazard ratio of death for infected patients with ineffective treatments ($\kappa_v$) | 1.04 | [47, 48] |
| Hazard rate of resistance development ($\varepsilon$) | 0.03 | [37, 38] |

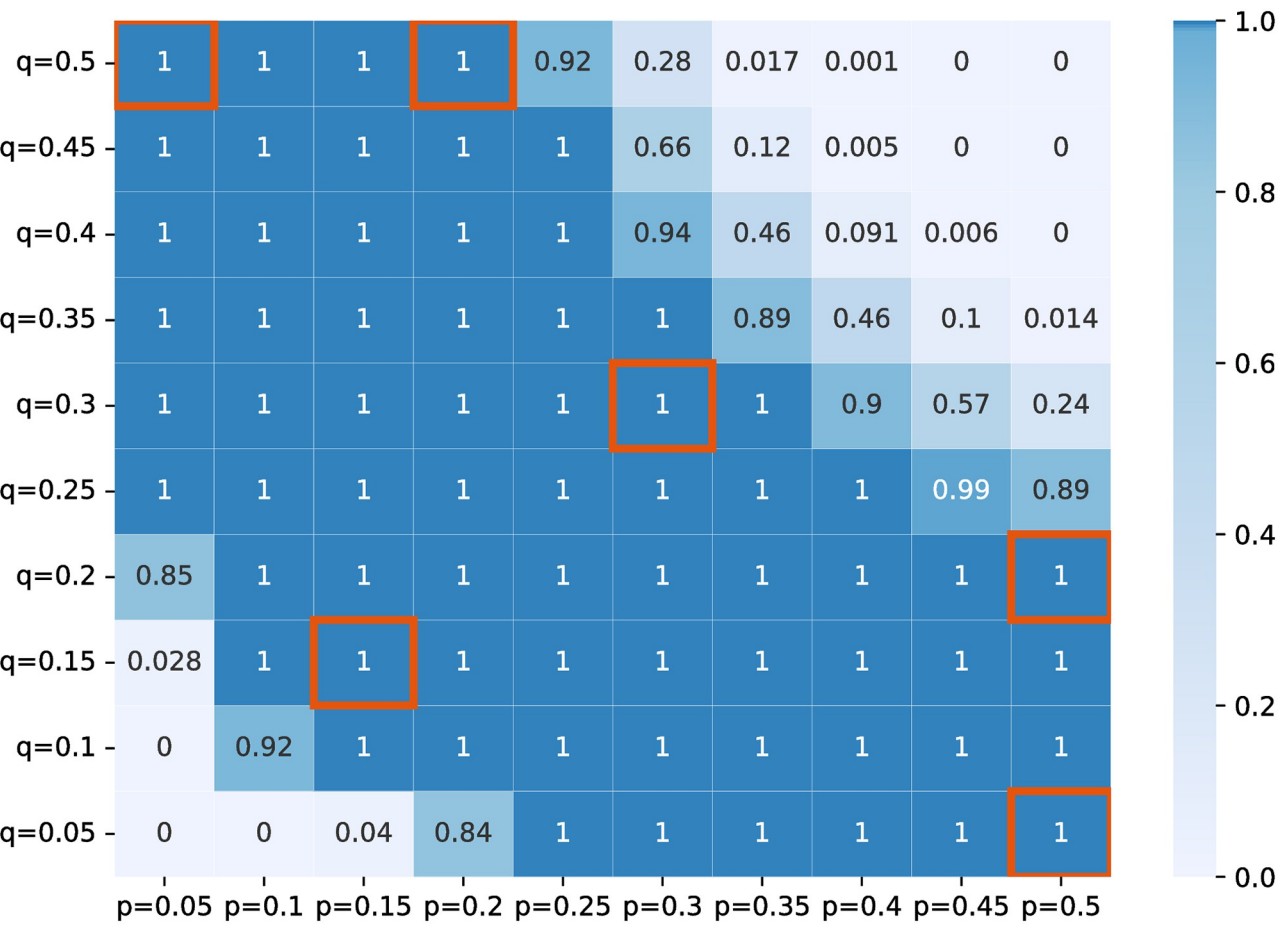

**Fig 3. Transmission probability calibration.** The number in each cell represents the fraction of all three outcomes (*i.e.* prevalence of resistance to CIP and TZP, and PA colonization) falling in the credible ranges over 1,000 experiments. We pick the transmission probability pairs marked in red borders for further simulations. *p* represents the probability of PA transmission from HCWs to patients upon each contact. *q* represents the probability of PA transmission from patients to HCWs during each visit.

conversions; (ii) incidence of nosocomial colonization by each PA strain; (iii) incidence of intrinsic mutations towards resistance against CIP, TZP, and both; (iv) administration of broad-spectrum antibiotics—this includes both definitive and correction use; (v) incidence of inappropriate empiric therapy either caused by resistance to TZP or strain conversions; (vi) incidence of super-infections; (vii) incidence of death. We illustrate the detailed calculation for the metrics of the outcome measurements in S1 Appendix.

## 2.4 Experiments and analyses

We simulate a two-arm RCT on ICUs with identical conditions, where half of the ICUs adopt de-escalation strategy (*i.e.* de-escalation group) and another half adopt continuation strategy (*i.e.* control group). The primary goal is to compare the measurement outcomes between ICUs adopting different strategies.

**Experiment 1:** *simulations with fixed parameter sets.* Firstly, we aim to investigate the theoretical benefits and trade-offs of de-escalation. We run a total of 1000 simulations for each fixed parameter set for both the de-escalation and continuation models and record all output

metrics at the end of each simulation day. In this way, we obtain 1000 data points to represent the distributions of all measurement values at the end of each simulation day for each study group. To visualize the trends of infection prevalence, we plot the average daily super-infections with 95% confidence interval in Fig 4 as a representative. To quantify the differences between the two study groups in their measurement distributions, we compute the *effect size* by calculating the Cohen's D value of each measurement distribution of the de-escalation and continuation model by the end of each week. Specifically, for each type of outcome measured by the end of any period in our case, the Cohen's D value, *d*, is calculated via the formula

$$d = \frac{\mu_{\text{DE}} - \mu_{\text{CT}}}{\sqrt{(\sigma_{\text{DE}}^2 + \sigma_{\text{CT}}^2)/2}},$$

where $\mu_{\text{DE}}$ and $\mu_{\text{CT}}$ are respectively the mean of the measurement distribution for the de-escalation and control group, and $\sigma_{\text{DE}}$ and $\sigma_{\text{CT}}$ are the standard deviations. We plot the variations of the effect sizes over time in Fig 5 for the high transmission mode with $p = q = 0.3$.

**Experiment 2:** *simulations with perturbed parameter values.* From a realistic point of view, the epidemiological conditions in ICUs differ from one to another. We thus perform sensitivity analyses to investigate the robustness of the theoretical benefits and trade-offs of de-escalation under perturbed parameter sets. For each transmission scenario, we perturb all model parameters in Table 1 around their fixed values. Specifically, to obtain a perturbed parameter set, we sample each model parameter from a normal distribution $\mathcal{N}(\mu, \sigma)$ with $\mu$ being the fixed parameter value and $\sigma$ being $0.1\mu$. Then we perform simulations based on 1000 perturbed parameter sets for each model and calculate the measurements as in Experiment 1. For a practical purpose, we calculate the number of arms and the length of study period needed to detect differences between the two study groups with results shown in Figs 6 and 7.

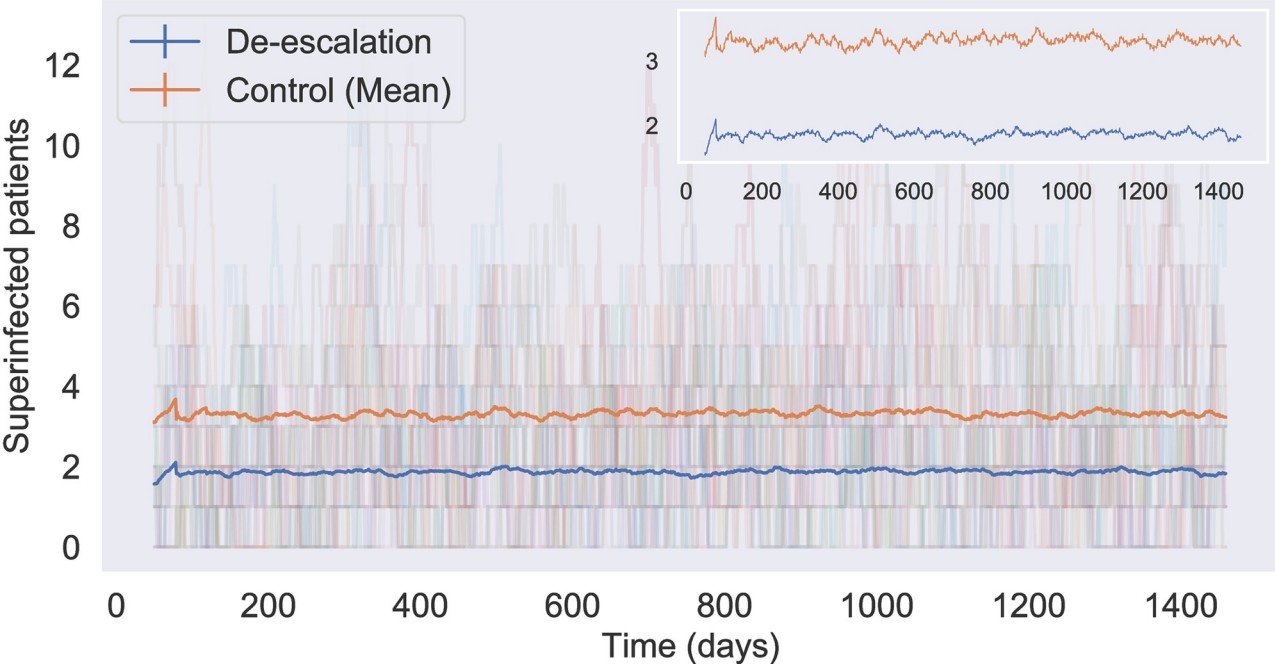

**Fig 4. Infection prevalence under high transmission mode ($p = q = 0.3$).** Each trajectory in the background light color tracks the average daily number of super-infected patients in a 64-bed ICU. The solid curves represent the mean value of daily super-infections for the de-escalation and control groups over 1000 simulations.

(a)

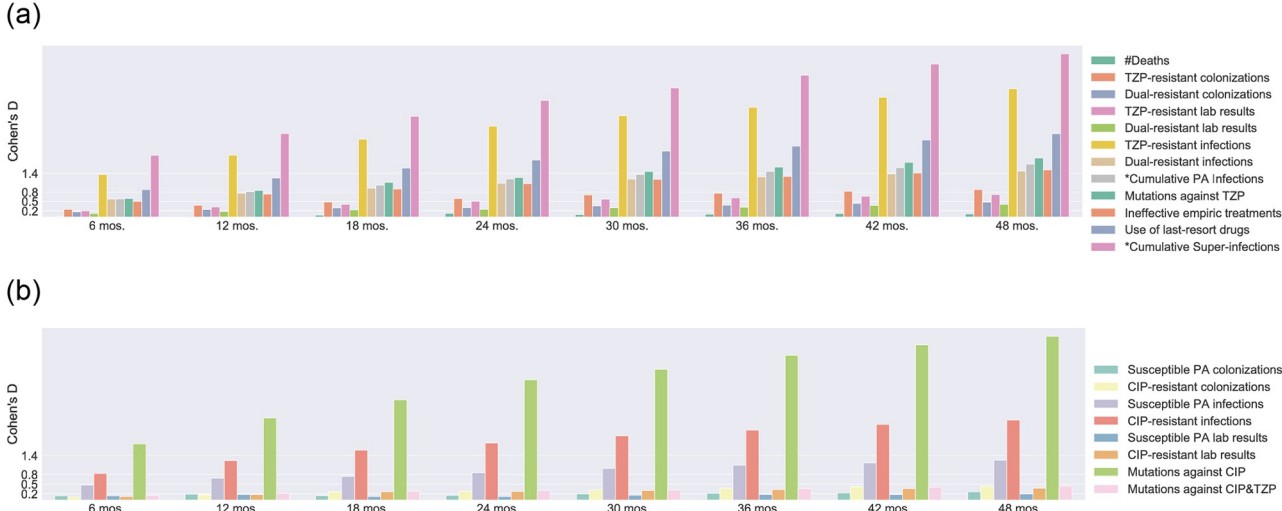

(b)

**Fig 5. Effect size of outcome measurements under high transmission mode ($p = q = 0.3$).** Each color bar represents the Cohen's D value of the corresponding outcome measurement between de-escalation and control groups measured at the end of each study period. Simulations are performed for each group for 1000 times with all parameter values fixed as in Table 1. The height of each bar represents the effect size between the distributions of de-escalation and control groups. Large effect size indicates high possibility for one to detect the projected difference in reality. Practically, effect size of 0.2, 0.5, 0.8, and 1.4 respectively correspond to 58%, 69%, 79%, and 92% probability of observing the control group under- or out- perform the mean of experimental group as projected. All measurements shown in (b) represent the trade-offs of de-escalation. All measurements except Deaths shown in (a) refer to the benefits. The use of TZP is a clear benefit of de-escalation with a significant Cohen's D value, so we omit this benefit in the figure.

## 3 Results

### 3.1 Theoretical effects of de-escalation

The theoretical benefits and trade-offs of de-escalation are based on the analysis of the synthetic data generated in Experiment 1. Our following conclusions are drawn from Figs 4 and 5.

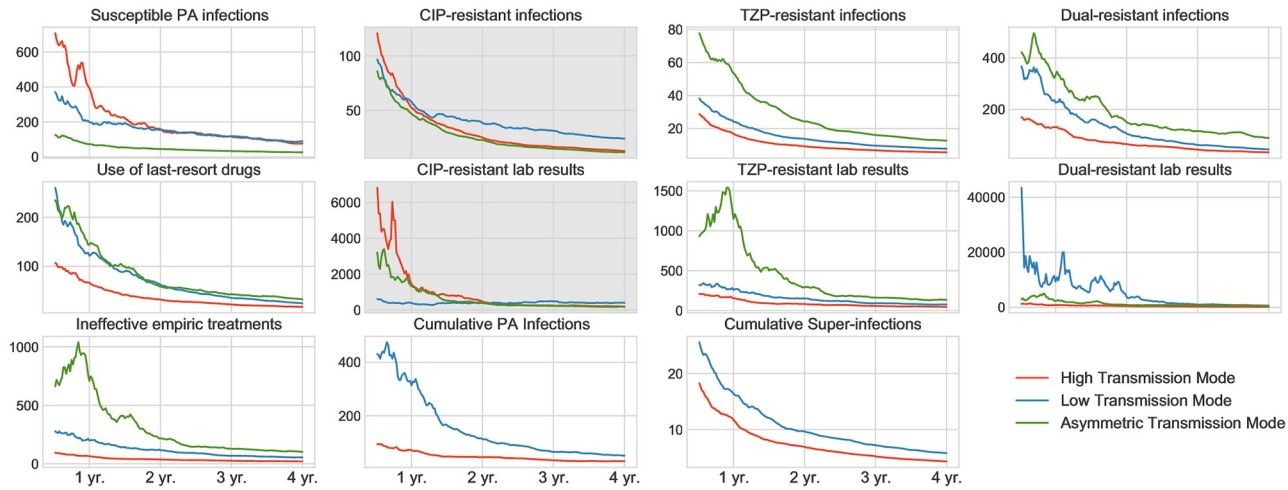

**Fig 6. Sample size estimation for RCTs with 16-bed ICUs under various transmission modes.** Each curve represents the number of balanced arm pairs needed to detect an expected difference between de-escalation and continuation groups in the corresponding measurement regarding the length of the study period (assuming 80% power, 5% type I error rate). Panels with white background refer to benefits of de-escalation, and those with gray background refer to trade-offs. High transmission mode refers to $p = q = 0.3$, low transmission mode refers to $p = q = 0.15$, and asymmetric transmission mode refers to $q = 0.5, p = 0.05$.

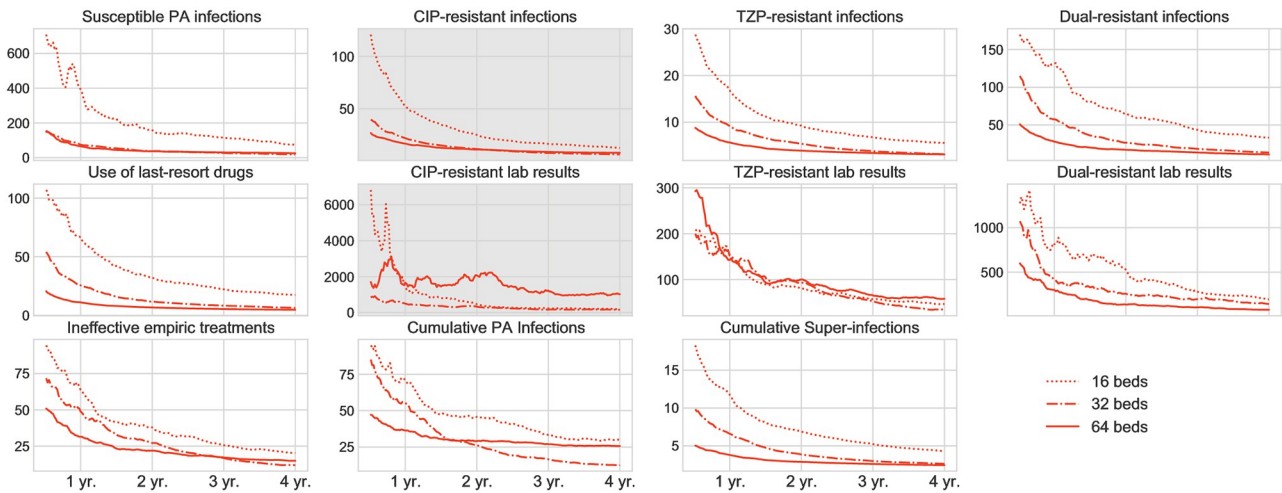

**Fig 7. Sample size estimation for RCTs under high transmission mode.** Each curve represents the number of balanced arm pairs needed to detect an expected difference between de-escalation and control groups in the corresponding measurement regarding the length of the study period (assuming 80% power, 5% type I error rate). Panels with white background refer to benefits of de-escalation, and those with gray background refer to trade-offs. Each curve type represents an RCT consisting of 16-bed, 32-bed, or 64-bed ICUs, where a 4:1 patient-HCW ratio is always maintained.

**Infection prevalence.** We plotted the prevalence of infections caused by all PA strains and the non-PA species for each study group under all transmission modes. Fig 4 serves as a representative of the same type of results, where the trajectories of every single run are shown in the background light colors. We assume that a certain fraction of patients is colonized by PA upon admission. So for most single runs, the daily infected patients (subject to all pathogens) may be zero at times but never extinct.

Under each transmission mode, the daily mean prevalence of infections caused by all strains stabilizes after two months of the experiment initiation for both groups, meaning that the incidence rates will eventually increase linearly. The stabilization of the daily mean prevalence also indicates that the impact of initial agent attributes on the model's outcomes will fade over time. Therefore, the differences between the two study groups, if any, can be detected after sufficiently long study periods.

As expected, de-escalation reduces the prevalence of resistance to TZP and other broader spectrum antibiotics, but in turn, increases the prevalence of the susceptible and CIP-resistant strains. Under high transmission mode, de-escalation demonstrates a clear advantage in reducing the prevalence of super-infections with daily mean case count differing by more than one between the two groups in 64-bed ICUs. However, such an advantage is almost negligible under low transmission mode.

**Benefits and trade-offs on incidence rates.** In practice, the effects of the two strategies would be measured and compared via incidence rates. In what follows, we quantify the difficulty/easiness of detecting the expected outcomes in RCTs. To do this, we compare the two strategies via the effect sizes between their distributions of each incidence rate measurement by the end of each study week.

We find most essential effects of de-escalation can be observed uniformly in all transmission intensities. *Benefits:* De-escalation possesses benefits in reducing infections caused by TZP- and dual-resistant strains, decreasing colonization caused by the TZP-resistant strain, limiting the use of TZP and the last-resort antibiotics, and avoiding inappropriate empiric therapies. De-escalation would also reduce the colonization of dual-resistant strain and the

mutations to TZP. But such effects are obvious in study periods with reasonable lengths. *Trade-offs:* De-escalation significantly increases the infection prevalence, colonization, and mutations associated with the CIP-resistant strain. De-escalation would also increase the infection prevalence and colonization of the susceptible PA strain and the mutation incidence toward dual-resistance but with a small difference from the control group. *Neutral Impacts:* Neither strategy possesses benefits in reducing mortality rates. Thus mortality rate is theoretically not affected by the underlying antibiotic use strategy and should not be considered measurements to compare study outcomes.

De-escalation's effects on cumulative PA infection and super-infection are not uniform under all transmission intensities. In all three high transmission intensities, de-escalation group demonstrates a strong reduction of super-infections and a moderate reduction of PA infections. Such benefits are still obvious in two low transmission intensities with $p = q = 0.15$ and $p = 0.5$, $q = 0.05$. However, in the low transmission scenario with $p = 0.05$, $q = 0.5$, such benefits are not observed. We conclude that de-escalation could reduce PA infections and super-infections under low HCW contamination probabilities, but such benefit is no longer valid under high HCW contamination probabilities. Therefore, good hand and medical instrument hygiene are vital not only in nosocomial infection prevention but also in achieving ideal outcomes when adopting antimicrobial de-escalation strategies.

In our following analyses, we focus on three specific transmission scenarios: **high transmission mode** with $p = q = 0.3$, **low transmission mode** with $p = q = 0.15$, and **asymmetric transmission mode** with $p = 0.05$, $q = 0.5$.

## 3.3 Practical detection of expected outcomes

In reality, ICUs with similar conditions would still differ slightly in model parameters. Thus we use the simulation results with perturbed parameter values as in Experiment 2 to infer the robustness of theoretical effects and the easiness of observing such effects in practice.

**Robustness of theoretical differences.** All theoretical benefits and trade-offs of de-escalation are robust under parameter perturbations. That is, each measurement continues to be a benefit or a trade-off under the analyses of data generated in Experiment 2. So RCTs with a sufficiently large sample size and long study period are likely to detect all the theoretical effects of de-escalation.

**Sample size and study period.** In Fig 6, we plot the sample size of study pairs needed to detect the projected difference between the de-escalation group and the control group for varying study lengths. Sample size is calculated by standard power analysis while we take statistical power ($\beta$, type II error) being 80% and significance ($\alpha$, type I error) being 5%. We omit the curves of cumulative PA infections and super-infections for the asymmetric transmission mode since these two measurements would theoretically establish negligible differences between the two study groups.

For RCTs with a study period of fewer than two years, a high transmission intensity environment would require fewer study arms to observe most of the de-escalation's benefits. But for RCTs with periods as long as four years, transmission intensities in the ICUs would not affect the number of required study arms. Therefore, the projected benefits and trade-offs of de-escalation can be ultimately observed if the following three conditions are satisfied: (i) all ICUs recruited for the study should share similar conditions in terms of the community resistance prevalence, patient types, hand hygiene standards, and transmission intensities; (ii) a sufficient number of study arms should be engaged; (iii) a sufficiently long study period should be planned.

We summarize the number of required study arms under a 4-year study period in Table 2. Based on our estimation, the number of study arms might be difficult to reach in reality. For example, the number of patients involved in prior RCTs on de-escalation ranged from 108 to 2,658 [12]. Whereas in our simulation for an RCT with 16-bed ICUs, 1,600 patients would be admitted in one single arm over four years. Therefore, insufficient study size and varying ICU conditions could contribute to the difficulties in detecting the effects of de-escalation in many prior clinical studies.

**Clinical and ecological resistance measurements.** There are significant differences in measuring the prevalence of resistant strains in hospitals. Clinical resistance data is routinely collected at empiric therapy and is analogous to the proportion of initial patient isolates resistant to CIP, TZP, or both. These correspond to our model's metrics CIP-resistant lab results, TZP-resistant lab results, and Dual-resistant lab results. However, the actual ecological measurement of infection prevalence should also consider patients secondarily infected by the specific strain. And these correspond to the metrics we use throughout our analyses for infection prevalence: CIP-resistant infections, TZP-resistant infections, and Dual-resistant infections. We calculate the clinical resistant measurements and investigate their robustness under parameter set perturbation. We first observe that the theoretical effects of de-escalation on the resistance prevalence are not affected by the measures adopted (Fig 5). Meaning the choice of measures would not alter the conclusions on the effects of de-escalation on the resistance

**Table 2. Table of measurements: Expectations and required study arms.** Required arms are obtained by taking the least possible sample size for each measurement at a four-year study period among 16-bed, 32-bed, and 64-bed ICUs.

| Measurement | Effect of De-escalation | Required Arms |
|---|---|---|
| Practically Measurable | | |
| CIP-resistant infections | Trade-off | $\sim 10$ |
| TZP-resistant infections | Benefit | $\sim 10$ |
| Dual-resistant infections | Benefit | $10 \sim 20$ |
| Cumulative PA infections | Benefit | $10 \sim 20$ |
| | Undetermined (asymmetric mode) | NA |
| Cumulative super-infections | Benefit | $\sim 10$ |
| | Undetermined (asymmetric mode) | NA |
| Use of TZP | Benefit | $<10$ |
| Use of last-resort drugs | Benefit | $\sim 10$ |
| Ineffective empiric treatments | Benefit | $\sim 12$ (high transmission intensities) |
| | | $30 \sim 60$ (low transmission intensities) |
| Deaths | Undetermined | NA |
| CIP-resistant lab results | Trade-off | $>300$ |
| TZP-resistant lab results | Benefit | $\sim 50$ |
| Dual-resistant lab results | Benefit | $>300$ |
| Practically Unmeasurable | | |
| CIP-resistant colonization | Trade-off | NA |
| TZP-resistant colonization | Benefit | NA |
| Dual-resistant colonization | Benefit | NA |
| Mutations against CIP | Trade-off | NA |
| Mutations against TZP | Benefit | NA |
| Mutations against CIP&TZP | Trade-off | NA |

Assuming 80% power and 5% type I error rate. Asymmetric mode refers to $p = 0.05$, $q = 0.5$. High transmission intensities refer to $p = q = 0.3$, $p = 0.2$, $q = 0.5$ and $p = 0.5$, $q = 0.2$. Low transmission intensities refer to $p = q = 0.15$, $p = 0.05$, $q = 0.5$ and $p = 0.5$, $q = 0.05$.

prevalence. However, the clinical resistance measurements are ineffective in detecting the differences between study groups. Fig 6 suggests that clinical resistance measurements would significantly scale up the demand of study arms to detect all projected effects. Therefore, inaccurate resistance measurement could also contribute to the failure in observing group differences of prior clinical studies.

**Size of ICUs.** Intuitively, the number of ICU beds would impact detecting differences between the experimental and control groups. A larger ICU size corresponds to more patient involvement during any time window. Then recruiting large-size ICUs may help shorten the study period and reduce the necessary study arms. Regarding this question, we perform simulations for 16-bed, 32-bed, and 64-bed ICUs while maintaining a 4:1 patient-HCW ratio. We repeat the simulations in Experiment 2 and analyze the required sample size in Fig 7. For measurements that are relatively easy to detect, increasing ICU size would, in general, help reduce the requested number of study arms. However, the benefit of large ICUs fades out if studies are already carried out for a long time (such as four years). Further, increasing ICU size would not change the difficulty of observing differences between study groups regarding the hard-to-detect measurements. And a considerable ICU size may not help reduce the requested study arms in significant scales.

## 4 Discussion

In this paper, we develop agent-based models to capture the complicated antibiotic treatment process, investigate the benefits and trade-offs of antimicrobial de-escalation on the prevalence of *P. aeruginosa* in ICUs, and infer the proper size of an RCT study for observing the desired outcomes. Difficulties in applying agent-based models lie in parameter calibration and the interpretation of model outputs. This work selects the transmission parameters by screening the model outputs via real-world data on infection prevalence. This method of calibration shares the same idea as in [31]. Our model outputs consist of multiple simulated cases from two distinct treatment models. To compare the outcomes, we utilized the basic statistical concept of Cohen's D value to quantify the differences between the outcome distributions of the two models. To inform the practical design of clinical trials, we further compute the sample size needed to detect the expected measurement differences, which shares similar thoughts as in [39].

We restrict our discussions on the patient-HCW ratio as 4:1, but our model codes are customized to simulate any patient-HCW ratio with any ICU size. Further, our model code is also valid to simulate any real-world study design with ICUs of various sizes. Many parameters and data used in this study are obtained from a diverse range of literature. In particular, the prevalence data for model calibration were obtained by researchers in Toronto teaching hospitals [14]. The hazard rate of resistance development is adopted from a previous estimation by the same group of researchers in Toronto [37]. The infection development probabilities for different bacterial species were parameterized from two diverse medical literature based on relatively small patient samples [40, 41]. Finally, the community level of PA resistance is provided by a study conducted in a hospital in Germany [36]. Ultimately, an ideal parameterization would request further clinical studies on many model parameters from a stable resource of patients and ICUs. One could easily adjust the parameter settings of our model to guide the design of future RCTs.

Our model provides a baseline analysis on the ecological effects of de-escalation on *P. aeruginosa*—a primarily nosocomial pathogen that is particularly adept at developing resistance [42–44]. By shortening the length of empiric therapy, the model can be easily modified to study the potential benefits of the future rapid diagnostic test on antimicrobial de-escalation

[5, 45]. It can also be re-parameterized to investigate the impacts of de-escalation on other bacterial species in ICUs. Due to the numerous bacterial species, strain types, and the intricate network on the drug-bug coverage, it is still hard to develop a more realistic model that considers the impacts of antimicrobial de-escalation on the overall ecology in ICUs. A complete model for antimicrobial stewardship should include nosocomial pathogens such as *methicillin-resistant Staphylococcus aureus* (MSRA), *vancomycin-resistant enterococci* (VRE), and *Clostridium difficile* (C. diff), and the corresponding detailed treatment protocols [46]. Specifically, *P. aeruginosa* could also be contaminated from water and environment, but this transmission route is not mechanistically modeled and only indirectly reflected in the parameters $a$, $m$, $r_1$, $r_2$. Incorporating the environmental contamination could further enhance our model thus our understandings of antimicrobial de-escalation. This work establishes a fundamental way and baseline computer codes for the analysis and simulation of future agent-based models with more realistic settings.

## Supporting information

**S1 Appendix. The model algorithm and outcome measurement can be found in the appendix file.**
(PDF)

## Author Contributions

**Conceptualization:** Xi Huo, Ping Liu.

**Formal analysis:** Xi Huo, Ping Liu.

**Investigation:** Xi Huo, Ping Liu.

**Methodology:** Xi Huo, Ping Liu.

**Software:** Ping Liu.

**Supervision:** Xi Huo.

**Validation:** Xi Huo, Ping Liu.

**Visualization:** Xi Huo, Ping Liu.

**Writing – original draft:** Xi Huo.

**Writing – review & editing:** Xi Huo, Ping Liu.

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
