## [Decision Letter · Decision Letter 0]

25 Aug 2023

PONE-D-22-29543An Agent-Based Model on Antimicrobial De-escalation in Intensive Care Units: Implications on Clinical Trial DesignPLOS ONE

Dear Dr. Huo,

Thank you for submitting your manuscript to PLOS ONE. After careful consideration, we feel that it has merit but does not fully meet PLOS ONE’s publication criteria as it currently stands. Therefore, we invite you to submit a revised version of the manuscript that addresses the points raised during the review process.

We look forward to receiving your revised manuscript.

Kind regards,

Junyuan Yang

Academic Editor

PLOS ONE

Journal Requirements:

XH was partially supported by the National Science Foundation (DMS-1853622 and DMS-2052648) and the College of Arts and Sciences at the University of Miami. This report is solely the responsibility of the authors and does not necessarily represent the official views of the National Science Foundation and the University of Miami.

XH was partially supported by the National Science Foundation (DMS-1853622 and DMS-2052648) and the College of Arts and Sciences at the University of Miami. This report is solely the responsibility of the authors and does not necessarily represent the official views of the National Science Foundation and the University of Miami.

Reviewers' comments:

Reviewer's Responses to Questions

**Comments to the Author**

1. Is the manuscript technically sound, and do the data support the conclusions?

Reviewer #1: Partly

2. Has the statistical analysis been performed appropriately and rigorously? 

Reviewer #1: I Don't Know

3. Have the authors made all data underlying the findings in their manuscript fully available?

Reviewer #1: No

4. Is the manuscript presented in an intelligible fashion and written in standard English?

Reviewer #1: Yes

5. Review Comments to the Author

Reviewer #1: 1. The P. aeruginosa may be not the major pathogen healthcare associated infections at present status in most hospitals, including America, Asia, and Europe regions.

2. The drug resistance of Pseudomonas spp. had several factors, including water and environment contaminations.

3. Needs more factors to confirm this hypothesis.

6. PLOS authors have the option to publish the peer review history of their article (what does this mean?). If published, this will include your full peer review and any attached files.

Reviewer #1: No

---

## [Author Response · Author response to Decision Letter 0]

9 Nov 2023

We appreciate the reviewer's efforts in reading our manuscript and providing the helpful insights. The concerns raised in 2 and 3 have helped us to enhance our knowledges about Pseudomonas, and have provided future directions of our model. Please find our detailed response to the reviewer's comments attached as a PDF file.

---

## [Decision Letter · Decision Letter 1]

1 Dec 2023

PONE-D-22-29543R1An Agent-Based Model on Antimicrobial De-escalation in Intensive Care Units: Implications on Clinical Trial DesignPLOS ONE

Dear Dr. Huo,

Thank you for submitting your manuscript to PLOS ONE. After careful consideration, we feel that it has merit but does not fully meet PLOS ONE’s publication criteria as it currently stands. Therefore, we invite you to submit a revised version of the manuscript that addresses the points raised during the review process.

We look forward to receiving your revised manuscript.

Kind regards,

Junyuan Yang

Academic Editor

PLOS ONE

Journal Requirements:

Reviewers' comments:

Reviewer's Responses to Questions

**Comments to the Author**

1. If the authors have adequately addressed your comments raised in a previous round of review and you feel that this manuscript is now acceptable for publication, you may indicate that here to bypass the “Comments to the Author” section, enter your conflict of interest statement in the “Confidential to Editor” section, and submit your "Accept" recommendation.

Reviewer #1: (No Response)

2. Is the manuscript technically sound, and do the data support the conclusions?

Reviewer #1: Yes

3. Has the statistical analysis been performed appropriately and rigorously? 

Reviewer #1: I Don't Know

4. Have the authors made all data underlying the findings in their manuscript fully available?

Reviewer #1: Yes

5. Is the manuscript presented in an intelligible fashion and written in standard English?

Reviewer #1: Yes

6. Review Comments to the Author

Reviewer #1: 1. P2 line 28-30, de-escalation of antibiotics focuses on susceptibility of pathogens (P. aeruginosa in this manuscript). The dual- drug resistance risk is much less than empirical therapy.

2. P3, line 87-89, ciprofloxacin (fluoroquinolones antibiotics) has high resistant rate in P. aeruginosa. Why do not consider ceftazidime as the de-scalation choice?

7. PLOS authors have the option to publish the peer review history of their article (what does this mean?). If published, this will include your full peer review and any attached files.

Reviewer #1: No

---

## [Author Response · Author response to Decision Letter 1]

12 Jan 2024

Response to Reviewers

Reviewer #1: 

1. P2 line 28-30, de-escalation of antibiotics focuses on susceptibility of pathogens (P. aeruginosa in this manuscript). The dual- drug resistance risk is much less than empirical therapy.

Response: we understand the reviewer’s question: since the definitive therapy is based on susceptibility profile of pathogens, then the dual-drug resistance risk in the definitive therapy stage is much less than its risk in the empirical therapy stage. However, we would like to emphasize that our main research question is to compare de-escalation vs. continuation, which are two drug use strategies (both involve empirical stage and definitive stage) only differed in their definitive therapy stages. And what we tried to express in the sentence mentioned is that it is intuitively uneasy to envision the benefit of de-escalation on the population-level prevalence of the dual-drug resistance. This is because de-escalation can lead to frequent use of both narrow-spectrum and broad-spectrum drugs, exposing the patient population to both drugs, and the selection pressure could trigger more spontaneous mutation thus result in higher dual-drug resistance prevalence. 

2. P3, line 87-89, ciprofloxacin (fluoroquinolones antibiotics) has high resistant rate in P. aeruginosa. Why do not consider ceftazidime as the de-scalation choice?

Response: we consider ciprofloxacin because it is the suggested drug for de-escalation used in Toronto teaching hospitals by their antimicrobial stewardship program. And in our prior collaboration with the physicians in Toronto (doi.org/10.1371/journal.pone.0171218), we used differential equation models to study the impacts of de-escalation under the same assumption. Further, comparing to ceftazidime (which is used to treat many infections such as bladder, bone, joint, etc.), ciprofloxacin is usually prescribed to treat an even wider variety of infections hence is used more often in ICUs. Further, the idea of de-escalation is to use antibiotics that have established high resistance prevalence whenever possible so that we can reduce the use of the most effective ones (such as pip-tazo in our model). Then the high resistant rate of ciprofloxacin qualifies it becoming a de-escalated option. In reality, there could be more than one de-escalated drug and different physicians may identify different de-escalation options. And considering multiple de-escalated drugs would be a great future direction of this study based on more realistic assumptions. We have added this explanation in the revised version.

We greatly appreciate the reviewer spending time reading our manuscript, as well as the helpful comments.

---

## [Editor Report · Decision Letter 2]

25 Mar 2024

An Agent-Based Model on Antimicrobial De-escalation in Intensive Care Units: Implications on Clinical Trial Design

PONE-D-22-29543R2

Dear Dr. Huo,

We’re pleased to inform you that your manuscript has been judged scientifically suitable for publication and will be formally accepted for publication once it meets all outstanding technical requirements.

Kind regards,

Junyuan Yang

Academic Editor

PLOS ONE
---

## [Editor Report · Acceptance letter]

5 Apr 2024

PONE-D-22-29543R2 

PLOS ONE

Dear Dr. Huo, 

I'm pleased to inform you that your manuscript has been deemed suitable for publication in PLOS ONE. Congratulations! Your manuscript is now being handed over to our production team.

Kind regards, 

on behalf of

Dr. Junyuan Yang 

Academic Editor

PLOS ONE